# Co-Inheritance of Heterozygous β⁰-Thalassemia with Single Functional α-Globin Gene: Challenges of Carrier Detection in Pre-Marital Screening Program for Thalassemia

**Hossein Jalali [1,2]** , **Hossein Karami [1]**, **Mohammad Reza Mahdavi [1,*]** and **Mehrad Mahdavi [2]**

1. Thalassemia Research Center, Hemoglobinopathies Institute, Mazandaran University of Medical Sciences, Sari 481538477, Iran
2. Sinaye Mehr Reseach Center, Mazandaran University, Sari 4817993375, Iran
* Correspondence: mahdavi899@gmail.com

**Abstract:** This is a report of a couple with abnormal hematological indices who were investigated for α & β-thalassemia mutations. Based on CBC and capillary hemoglobin electrophoresis results, the male and female subjects were β & α-thalassemia carriers, respectively. Multiplex-Gap-PCR and Sanger sequencing techniques were used for the identification of mutations on α and β-globin genes. The DNA test showed the presence of c.315 + 1 G > A mutation on β-globin gene of male subject while the female case had – ᴹᴱᴰ double gene deletion and c.427T > C mutation on α-globin and, interestingly, she was also a carrier for c.315 + 1 G > A mutation on β-globin gene. Cases with the coinheritance of heterozygous β⁰-thalassemia with one functional α-globin gene have normal HbA2 levels that may lead to their being misdiagnosed as β-thalassemia carriers, especially in premarital screening programs for thalassemia. Therefore, β-globin gene sequencing is recommended in cases with normal Hb electrophoresis and reduced hematological indices in premarital screening programs for thalassemia, especially in regions with a high frequency of β-globin mutations, in order to identify all the β-thalassemia carriers.

**Keywords:** β-thalassemia; α-thalassemia; premarital screening program





## 1. Introduction

α and β-Thalassemia are among the most frequent genetic disorders in the world. The diseases are caused by the mainly reduced synthesis of α and β-globin genes [1]. The level of globin chains imbalance that underlies ineffective erythropoiesis is the main factor that affects the phenotype of the patients [2].

Thalassemia is a common health problem in Mediterranean region, the Middle East, China, Southeast Asia, and West Africa [3]. In northern Iran, at least 15 and 10% of the population are α and β-thalassemia carriers, respectively [4,5]. Due to the high frequency of β-thalassemia carriers in Iran, a national pre-marital screening program for the identification of β-thalassemia carriers began in 1991 [6].

Typically, gene deletions in α-thalassemia carriers result in a low mean corpuscular volume (MCV), low mean corpuscular hemoglobin concentration (MCH) and normal values of hemoglobin A2 (<3.5%), while carriers of β-thalassemia show low hematological indices that are accompanied by elevated levels of HbA2 (≥3.5%) [1]. HbA2 levels may not increase in cases with coinheritance of α&β-thalassemia, which leads to the misdiagnosis of carriers in premarital screening programs [7,8].

The current study aimed to report on the co-inheritance of α- and β-thalassemia mutations in a female case with normal HbA2 levels, while her wife was a β-thalassemia carrier. Misdiagnosis of this case could have led to childbirth with thalassemia.

## 2. Case Presentation

As a part of a premarital screening program for thalassemia, a young couple was referred to the Fajr Medical genetics and pathobiology lab. They signed a written informed consent and agreed to attend the study. As a primary step, complete blood count (CBC) and capillary hemoglobin electrophoresis (Sebia, France) were applied (Table 1). The results indicated that the male and female subjects were β & α-thalassemia carriers, respectively (Table 1). Since both of them had reduced hematological indices, molecular analysis was applied for the detection of α & β-globin mutations.

**Table 1.** Hematological indices of the couple referred for premarital screening of the β-thalassemia.

|  | Age (y) | RBC ($\times 10^6$/μL) | Hb (g/dL) | MCV (fl) | MCH (pg) | Hb-A (%) | Hb-A2 (%) | Hb-F (%) | Other Hb (%) |
|---|---|---|---|---|---|---|---|---|---|
| Male | 27 | 6.1 | 12 | 63.1 | 19.4 | 94.5 | 5.1 | >0.5 | - |
| Female | 23 | 6.8 | 10.3 | 46.9 | 15.1 | 93.8 | 3.2 | 2.2 | 0.8 (HbCS) |
| Reference Range | - | 4.5–6.3 | 12–16 | 80–95 | 27–32 | 94.5–98 | <3.5 | <2 | - |

In the next step, for molecular analysis, genomic DNA was extracted from peripheral blood using QIAamp DNA Mini Kit (Qiagen, Germany) and, for the detection of common α-globin gene deletions (-α3.7, -α4.2.--MED, --20.5, -- SEA, and -- FIL), the multiplex gap-PCR method was used [9]. For the identification of α and β-globin point mutations, the entire α and β-globin genes were sequenced using the direct DNA sequencing approach (3130XL, ABI, USA). The molecular analysis confirmed that the male subject is a β-thalassemia carrier with c.315 + 1 G mutation (NM_000518.5), which is a common mutation in the region. He had no α-globin mutation.

Investigation of α-globin gene mutations on female case showed only one functional α-globin gene: actually, she carried --Med double gene deletion and c.427T > C mutation (NM_000517.6) or Hb Constant Spring simultaneously (Figure 1). β-globin gene analysis showed that she is also a β-thalassemia carrier (Heterozygote for c.315 + 1 G > A mutation) (Figure 1). Although the CBC and capillary electrophoresis indicated that the female is an α-thalassemia carrier, the sequencing results showed that the case also carries the β- globin gene mutation and she could have been misdiagnosed if only the CBC and electrophoresis test results were considered for premarital screening. The identified mutations were also confirmed in her parents. She had a normal phenotype with no symptoms of thalassemia. Since the male and female subjects were β-thalassemia carriers, prenatal diagnosis of β-thalassemia was recommended for this couple.

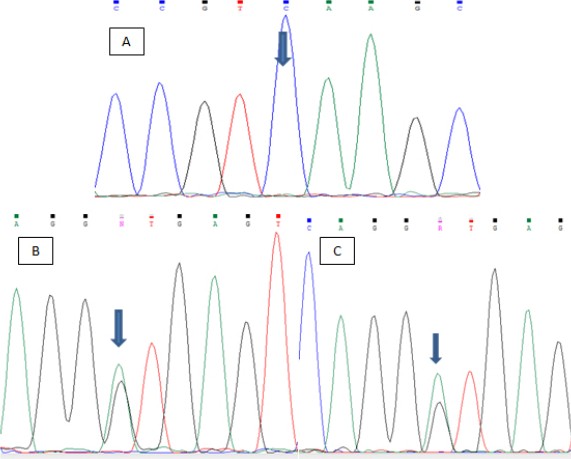

**Figure 1.** The PCR sequencing analysis of the cases: (**A**) The Hb Constant Spring mutation on α-globin gene of female case. (**B**) The position of c.315 + 1 G > A mutation on male case (**C**) The position of c.315 + 1 G > A mutation on female subject.

## 3. Discussion

β-thalassemia is the most common inherited disease in Iran, especially in the southern and northern provinces (more than 10 % carrier frequency), [10–12]. A significant reduction was observed in the emergence of new cases in most regions of the country as the pre-marital carrier screening programs for thalassemia began in 1991, [6]. However, there are still rare new childbirths with thalassemia due to failures in the screening program [13,14]. The screening program is primarily based on hematological indices and hemoglobin electrophoresis test results. The carriers of β-thalassemia usually show low hematological indices, accompanied by higher levels of HbA2 ($\geq$3.5%). However, there are some carriers of β-thalassemia, like the presented case, that have normal HbA2 levels, and misdiagnosis may lead to childbirth with thalassemia.

A similar condition was reported in a couple from China, where a single functional $\alpha$-globin in combination with β-globin mutation was detected in a man and his wife was a β-thalassemia carrier [15]. The husband had $--^{SEA}/\alpha^{CS}\alpha$ genotype for $\alpha$-thalassemia and he was heterozygote for codons 41–42 (-TCTT) mutation on β-globin gene and his wife was heterozygote for IVS-II-654 (C > T) mutation. Similar to the presented female case, although the husband was heterozygote for β-globin gene mutation, his Hb A2 levels were in the normal range (2.8%).

The co-existence of $\alpha$ and β-thalassemia is not very rare in the region; however, in the presented case, due to the presence of just one functional $\alpha$-globin gene, the HbA2 levels were reduced to the normal rang. Misdiagnosis of these cases increases the risk of childbirth with thalassemia. Moreover, there are some silent beta thalassemia mutations that do not change the haematological indices and can be missed during screening programs [16]. The rapid and precise screening of carriers is vital in the population with high thalassemia incidence rates to prevent the misdiagnosis of carriers, and the presentation of complicated cases helps to get comprehensive knowledge about carrier detection.

The present case indicates that coexistence of $\alpha$ and β-globin gene defects may lead to misdiagnosis of β-thalassemia carriers. Therefore, β-globin gene sequencing is recommended in cases with normal Hb electrophoresis and reduced hematological indices in premarital screening program for thalassemia, especially in regions with a high frequency of β-globin mutations in order to identify all of the β-thalassemia carriers.

**Author Contributions:** M.M. and H.J.; methodology, M.M.; software, H.K.; formal analysis, M.M. and H.K.; data curation, H.J. and H.K.; writing—original draft preparation, M.R.M., writing—review and editing, M.R.M.; project administration. All authors have read and agreed to the published version of the manuscript.

**Funding:** This research received no extra funding.

**Institutional Review Board Statement:** Not applicable.

**Informed Consent Statement:** Not applicable.

**Data Availability Statement:** Data can be made available upon request to the corresponding author.

**Acknowledgments:** The authors are grateful to Bita Talebi, Fatemeh Alizadeh, and Maryam Rahimi for their assistance in sampling and writing of the manuscript. This case report was coordinated by Fajr Genetics and Pathobiology Laboratory.

**Conflicts of Interest:** The authors report no conflict of interest. The authors alone are responsible for the content and writing of this article.

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
