# Peer review of "Co-Inheritance of Heterozygous β0-Thalassemia with Single Functional α-Globin Gene: Challenges of Carrier Detection in Pre-Marital Screening Program for Thalassemia"

_thalassrep, doi:10.3390/thalassrep12030015_

Round 1

Reviewer 1 Report

The authors presented an interesting case that is quite common in populations with high prevalence of thalassaemias. The importance of the case is not demonstrated accordingly and it should be discussed further. 

Figure 1: The Accession no of sequence used should be presented.

In general, all the sections of the case need improvement. 

Especially the discussion is very short and the importance of the case is poorly presented and not justified properly. The other case that is mentioned in the Discussion is irrelevant while the link to the one presented is not clear. 

Author Response

Dear Reviewer 

Thanks for your helpful comments. The Article is revised on your comments.

All parts of the article is changed especially the case presentation and discussion. The accession number of the sequence is also presented in the text.

Regards

Reviewer 2 Report

Review of article entitled: Co-inheritance of heterozygous β0-thalassemia with single func-tional α-globin gene: challenges of carrier detection in pre-marital screening program for thalassemia: a case report, Mazandaran, Iran.

The authors describe an interesting case of a female with co-inheritance of a-thalassemia and a b-globin gene mutation, diagnosed due to national screening program. 

Discussion though could be more extensive. I would like to know if it is a routine in your country to look always for mutations when electrophoresis and hematological parameters are compatible with a or b thalassemia carrier? or was there something else in this case that lead to molecular analysis (for example pregnancy)?

Methods

Complete blood count (CBC) and capillary haemoglobin electrophoresis (Sebia, France) were applied (Table1).

Table 1 belongs to results 

Did you check also ferritin?

Results

How exactly were mutations shared to her parents ? 

Discussion

Too short

Reference

1.     Not so recent

References 1,2,7,8,10: year of publication is missi

Author Response

Dear Reviewer 

Thanks for your helpful comments. The Manuscript has changed based on your comments.

Regards

Round 2

Reviewer 1 Report

Some improvement has been observed, however, not enough to render this manuscript publishable in its current form. 

Author Response

Dear Reviewer 

thanks for your comments

The article is revised based on your comments. The changes are indicated in Blue. 

Regards 
